# Anti-Inflammatory Effect of Fucoidan from *Costaria costata* Inhibited Lipopolysaccharide-Induced Inflammation in Mice

**DOI:** 10.3390/md22090401

**Published:** 2024-09-02

**Authors:** Wei Zhang, Peter C. W. Lee, Jun-O Jin

**Affiliations:** 1Shanghai Public Health Clinical Center, Shanghai Medical College, Fudan University, Shanghai 201508, China; weiwei061215@126.com; 2Department of Biochemistry and Molecular Biology, Brain Korea 21 Project, Asan Medical Center, University of Ulsan College of Medicine, Seoul 05505, Republic of Korea; 3Department of Microbiology, Brain Korea 21 Project, Asan Medical Center, University of Ulsan College of Medicine, Seoul 05505, Republic of Korea

**Keywords:** fucoidan, *costaria costata*, anti-inflammation, dendritic cells, health supplement

## Abstract

Seaweed extracts, especially fucoidan, are well known for their immune-modulating abilities. In this current study, we extracted fucoidan from *Costaria costata*, a seaweed commonly found in coastal Asia, and examined its anti-inflammatory effect. Fucoidan was extracted from dried *C. costata* (FCC) using an alcohol extraction method at an extraction rate of 4.5 ± 0.21%. The extracted FCC comprised the highest proportion of carbohydrates, along with sulfate and uronic acid. The immune regulatory effect of FCC was examined using bone marrow-derived dendritic cells (BMDCs). Pretreatment with FCC dose-dependently decreased the lipopolysaccharide (LPS)-induced upregulation of co-stimulatory molecules and major histocompatibility complex. In addition, FCC prevented morphological changes in LPS-induced BMDCs. Moreover, treatment of LPS-induced BMDCs with FCC suppressed the secretion of pro-inflammatory cytokines. In C57BL/6 mice, oral administration of FCC suppressed LPS-induced lung inflammation, reducing the secretion of pro-inflammatory cytokines in the bronchoalveolar lavage fluid. Finally, the administration of FCC suppressed LPS-induced sepsis. Therefore, FCC could be developed as a health supplement based on the observed anti-inflammatory effects.

## 1. Introduction

Plant-derived natural extracts are known to possess diverse biological activities [1,2,3]. In Asia, seaweed has long been recognized to exhibit various functions [4,5]. In traditional Chinese medicine, seaweed is widely known to exert biological activities, such as anticancer effects, and is available as pills for consumption [6]. Among brown algae, *Fucus vesiculosus* (*F. vesiculosus*), *Undaria pinnatifida*, and *Costaria costata*, which are widely distributed along the coasts of China and Asia, have been used in traditional Chinese medicine [7,8,9].

In seaweed, fucoidan, laminarin, and porphyrin are identified as representative polysaccharides [10]. These are all well known to exert biological activities [5]. Fucoidan is a polysaccharide contained in brown algae and is the main component of the sticky substance on the surface of brown algae [11]. Fucoidan is a sulfated polysaccharide comprising L-fucose [7,12]. Fucoidan extracted from *F. vesiculosus* is actively being explored and has been confirmed to exert anticancer, anti-inflammatory, and antioxidant effects [13,14,15]. In particular, fucoidan has shown an inhibitory effect of cyclooxygenase-1 and 2 (COX-1/2), mitogen-activated protein kinase p38, hyaluronidase, protein denaturation, and RBC membrane stabilization [16,17,18,19]. In addition, the formulation and optimization of a fucoidan-based cream with anti-inflammatory properties also resulted in dose-dependent inhibition of carrageenan-induced edema and relief of mechanical pain in rats [20].

Inflammation is an immune system response to treat infection. However, when activated indiscriminately, inflammation causes damage to healthy tissues [21]. Inflammation occurs owing to the activation of innate immune cells and causes tissue damage via the expression of pro-inflammatory cytokines [22]. Representative innate immune cells involved in inflammatory responses include macrophages, dendritic cells (DCs), and neutrophils. Among them, macrophages and DCs in tissues are activated to remove invading antigens [23]. During this process, inflammatory cytokines are expressed and eliminate infectious pathogens; however, these cytokines also damage healthy tissues [24]. Additionally, components of pathogens, such as lipopolysaccharide (LPS), induce immune activation and cause inflammation [25]. Therefore, appropriate immune regulation is required following the elimination of pathogens, which can suppress damage to healthy tissues [26].

DCs are the first cells to respond to pathogens that have invaded peripheral tissues, and they worsen inflammation by inducing the migration of neutrophils to infected tissues via the secretion of inflammatory cytokines [27]. DC activation is indicated by the presence of various indicators. Activated DCs overexpress the co-stimulatory molecules CD80 and CD86 on their surface [28]. DCs bind to CD28 on T cells and induce T cell activation. Additionally, activated T cells express CD40 ligand, which stimulates the CD40 overexpression on DCs, leading to further activation [29]. DCs overexpress class I and class II major histocompatibility complex (MHC) to present captured antigens [27]. Furthermore, DCs secrete pro-inflammatory cytokines such as interleukin (IL)-1β, IL-6, IL-12, and tumor necrosis factor (TNF)-α, thereby inducing inflammation [27]. Therefore, various drugs capable of suppressing DC activation are under development to prevent the worsening of inflammation [30].

LPS is a substance present in the cell membrane of *Escherichia coli* (*E. coli*) and is known as an endotoxin [31]. The endotoxin secreted by *E. coli* that invades the body can cause acute pneumonia and induce sepsis due to cytokine storm [32]. Given that humans are 1000 to 10,000 times more sensitive to endotoxins than mice, various therapeutic agents and health supplements are under investigation to treat inflammation and sepsis caused by endotoxins [33]. *C. costata* is a brown alga abundant in Asia and China. As most Asians already consume *C. costata* as a food source, it can be obtained at a low price [8]. *C. costata* has been used as a medicinal herb in Chinese medicine. However, the medicinal function of fucoidan, the main component of *C. costata*, remains poorly established. In particular, fucoidan exhibits anti-inflammatory or immune-activating effects. Therefore, in this study, the immune regulatory function of fucoidan from *C. costata* (FCC) was investigated. We explored the effect of FCC on LPS-stimulated BMDCs in an in vitro study. Furthermore, we evaluated the anti-inflammatory effects of FCC in mice challenged with LPS-induced lung inflammation and sepsis.

## 2. Results

### 2.1. Isolation and Characterization of FCC

We extracted FCC using a previously known method (Figure 1). Herein, we extracted 2.25 ± 0.10 g of FCC from 50 g of *C. costata*, which represents an extraction yield of 4.5 ± 0.21%. Although there was no significant difference in extraction yield when compared with that reported previously, the relatively large error between extractions could be due to variables such as seaweed harvest time and region.

Based on Fourier transform infrared (FT-IR) data, FCC had a band at 3425 cm^−1^, which corresponded to the O-H stretching vibration of carbohydrates. Additionally, a band at 2923 cm^−1^ representing the C-6 group of fucose was confirmed. We confirmed the presence of bands such as O-H, S=O, and C-O-S, indicating that the FT-IR results of the extracted FCC were similar to those of other fucoidans (Figure 2). The molecular weight of FCC was found to be 420 KDa.

The monosaccharide composition of FCC is shown in Table 1. Fucose was identified as the main constituent monosaccharide in fucoidan, along with xylose, mannose, glucose, and galactose. The FCC isolated in this study contained 25.4% of fucose, along with 5.2% of mannose, 4.1% of galactose, 3.1% glucose, and 2.3% of xylose. These content ratios did not differ substantially from those in fucoidan from *F. vesiculosus* (FFV). Nevertheless, FCC had a relatively lower fucose content and higher mannose content than FFV. Based on these results, we determined that fucoidan was appropriately extracted from *C. costata*. We next examined the biological activity of FCC.

### 2.2. FCC Inhibited the Activation of Bone Marrow-Derived DCs (BMDCs)

The biological effect of FCC was evaluated using the BMDCs. Following pretreatment with FCC for 1 h, BMDCs were incubated with LPS for 24 h. Treatment with LPS alone resulted in BMDCs with long extending branches. Treatment with FCC alone did not alter the BMDC morphology when compared with phosphate-buffered saline (PBS) treatment (Figure 3A). Pretreatment with FCC markedly reduced the LPS-induced changes in the branching morphology of BMDCs (Figure 3A). Treatment with LPS increased the expression of active surface markers of BMDCs, and pretreatment with FCC tended to suppress their expression in a concentration-dependent manner (Figure 3B).

### 2.3. FCC Inhibited Pro-Inflammatory Cytokine Secretion in LPS-Stimulated BMDCs

Next, we examined the secretion of pro-inflammatory cytokines by BMDCs. Treatment of BMDCs with FCC alone did not increase the levels of IL-1β, IL-6, IL-12, and TNF-α in the culture medium, indicating that FCC did not induce the activation of BMDCs. Treatment with LPS rapidly increased the concentrations of the examined pro-inflammatory cytokines in the BMDC culture medium. Pretreatment with FCC at different concentrations significantly reduced the LPS-induced secretion of pro-inflammatory cytokines (Figure 4). Therefore, these findings suggested that FCC could suppress the LPS-induced secretion of pro-inflammatory cytokines.

### 2.4. FCC Suppressed LPS-Induced Acute Lung Inflammation

To examine the anti-inflammatory effect of FCC in mice in vivo, mice were administered FCC at 24 h intervals for 3 days. Six hours after the last FCC administration, mice were administered LPS to induce lung inflammation. Although treatment with FCC alone did not induce any signs of inflammation in the lungs of treated mice, LPS administration induced the infiltration of leukocytes (Figure 5A). Furthermore, treatment with FCC reduced the degree of LPS-induced inflammation in treated mice (Figure 5A). Treatment with FCC dramatically reduced the LPS-induced pro-inflammatory cytokine secretion in the bronchoalveolar lavage (BAL) fluid. These results demonstrated that FCC could suppress inflammatory lung disease caused by LPS.

### 2.5. Oral Administration of FCC-Suppressed LPS-Induced Sepsis in Mice

Next, we examined whether FCC could attenuate LPS-induced sepsis. Mice were orally administered 25, 50, and 100 mg/kg FCC daily for 3 days, with LPS (10 mg/kg) administered intraperitoneally 24 h after the last FCC administration. All mice died within 42 h of LPS administration (Figure 6A). In the group treated with 50 mg/kg FCC, 40% of the mice survived, and in the group treated with 100 mg/kg FCC, 50% of the mice survived (Figure 6A). Serum concentrations of alanine transaminase (ALT) and aspartate transaminase (AST), hepatocyte enzymes that indicate increased liver toxicity induced by LPS, were significantly reduced in FCC-treated mice (Figure 6B). Moreover, pretreatment with FCC reduced the LPS-induced increase in pro-inflammatory cytokines in mice (Figure 6C). The anti-inflammatory effect of FCC against LPS was comparable to that of the positive control, dexamethasone (Dex). Accordingly, oral administration of FCC could suppress LPS-induced sepsis.

## 3. Discussion

Although extracted natural products may be developed into pharmaceuticals, most are likely to be employed as health supplements [36]. Considering medicinal herbs derived from Chinese medicine, it is necessary to establish proof of efficacy upon oral administration [37]. In this study, we confirmed that orally administered FCC can inhibit LPS-induced pneumonia and sepsis. Additionally, FCC did not induce any adverse effects, such as inflammation or hepatotoxicity, in healthy mice. Therefore, the FCC could be developed as a supplement to improve health.

Natural polysaccharides, including fucoidan, may have different ratios of monosaccharides depending on the extraction method and harvest time of the raw material. In particular, in the case of fucoidan, several papers have analyzed and reported its monosaccharide components, but there are some differences [38,39,40,41]. The FCC extracted in this study also shows some differences from the ratio of monosaccharide components shown in previous studies [8,42]. The biological effects of fucoidan often showed differences, which may be due to the composition ratio of monosaccharides. Therefore, it is necessary to secure FCC with a similar monosaccharide ratio, as in previous studies, and compare the anti-inflammatory effect with the FCC of this study to confirm the biological activity according to the monosaccharide ratio.

In addition, whether fucoidans exert an immune-activating or an immune-suppressing effect remains debatable. FFV, the most studied fucoidan, is known to induce the activation of DCs and macrophages [41,43]. Moreover, FFV reportedly exerts anti-inflammatory effects in colitis [44]. These differences may be due to differences in the composition ratio of monosaccharides, but differences in experimental methods to confirm efficacy may also be the reason. In the case of fucoidans capable of inducing immune activity, most were administered intraperitoneally or intravenously, whereas those with immunosuppressive effects were administered orally [45]. In this current study, orally administered FCC suppressed LPS-induced pneumonia and sepsis. Therefore, our findings are consistent with those of previous studies, which reported the anti-inflammatory function of other orally administered fucoidans.

Upon activation and maturation, DCs are well known to induce differentiation and activity of T cells [28]. Mature DCs overexpress active cofactors and MHC molecules on their surface, resulting in T cell activation. Active cofactors include CD40, CD80, and CD86, which bind to CD28 on T cells and induce T cell proliferation [30]. These active cofactors in DCs are crucial for inducing T cell activity but are used to treat various diseases by suppressing their expression or blocking their binding [46]. The low expression of active cofactors can induce immune tolerance and protect normal tissues [47]. FCC inhibited the expression of MHC molecules and LPS-induced activation of cofactors. DC utilizes MHC for antigen presentation, and MHC is used to efficiently deliver foreign antigens to T cells [48]. However, there are concerns that autoantigens may induce autoimmune and inflammatory diseases due to overexpression by MHC molecules; thus, it is necessary to modulate expression [49]. In this study, treatment with FCC efficiently suppressed LPS-induced BMDC activity. Therefore, FCC-mediated inhibition of active cofactors and MHC molecules could be harnessed to treat inflammatory diseases.

Pro-inflammatory cytokines are indicators of inflammatory diseases and major factors that damage healthy tissues [24]. Pro-inflammatory cytokines inherently cause damage to healthy tissues and can worsen inflammation by inducing the activity of other immune cells [22]. IL-1β and TNF-α are major toxic substances in the body, causing fever or cell necrosis [26,50]. Therefore, various therapeutic agents capable of modulating these actions are being developed. Natural products, such as fucoidan, have been shown to exert anti-inflammatory effects [7,44]. However, the anti-inflammatory effects of FCC remain poorly explored. Herein, we demonstrated that FCC could inhibit the expression of pro-inflammatory cytokines.

LPS is a cell membrane component of gram-negative bacteria known as endotoxin and is known to induce severe immune activation [31]. Humans are 1000–10,000 times more sensitive to LPS than mice [33]. Exposure to LPS induces the expression of large amounts of pro-inflammatory cytokines, which induce a cytokine storm, leading to sepsis [32]. Sepsis is the first to damage the functioning of the lungs and liver and can even lead to death [51]. Given that inflammatory cytokines secreted from the lungs and liver can directly induce damage, appropriate control is necessary [51]. In this study, treatment with FCC substantially reduced the LPS-mediated secretion of pro-inflammatory cytokines into the BAL fluid of mice. In addition, serum concentrations of cytokines in LPS-induced septic mice were reduced following pretreatment with FCC. These data indicate that FCC can exert an anti-inflammatory effect even when administered orally. Although various food-derived substances under development as health supplements frequently exhibit excellent effects in vitro, confirming their in vivo efficacy can be challenging. In addition, the effectiveness of these substances is often verified by intraperitoneal or intravenous administration, which is difficult to apply in human participants, and the research results are insufficient for development as supplements for health benefits. However, in this study, the anti-inflammatory properties of FCC were observed upon oral intake in the mice. Thus, these results underscore the possibility of developing FCC as a health supplement capable of exerting anti-inflammatory effects.

In this study, the anti-inflammatory effect of FCC was investigated. Most adults are frequently exposed to inflammatory diseases. For this purpose, anti-inflammatory drugs with various ingredients have been developed and are currently being used on patients [52]. Most of these are synthetic drugs and can cause various side effects on the body [53,54]. Since FCC is the main component of seaweed commonly consumed in Asia, it is expected to have very low side effects on the body. Therefore, it is expected that FCC can be developed into an alternative medicine or functional food with alternative anti-inflammatory efficacy based on more detailed component analysis and a high extraction rate.

## 4. Materials and Methods

### 4.1. Collection of C. costata

*C. costata* was collected from the coast of Dalian, China, in April–May 2023 and washed five times with sterile filtered water to remove sand and necrotic parts. Subsequently, *C. costata* was dried at room temperature for two weeks. FCC was identified by Professor SangGuan You of the Department of Marine Food Science and Technology, Gangneung-Wonju National University in Korea, in accordance with the relevant literature reported by Wang and Song [8]. FCC specimens are stored at the Shanghai Public Health Clinical Center (SPH-DA-1).

### 4.2. Isolation of FCC

FCC was extracted with a minor modification, based on previous studies, known as the fucoidan extraction method [55]. As detailed in Figure 1, dried *C. costata* was ground into powder using an electric grinder. Then, 50 g of crushed *C. costata* was suspended in 1 L of 85% ethanol and stirred at room temperature for 12 h. Subsequently, protein components were removed using centrifugation at 2000 rpm for 10 min. The remaining material was dried again at room temperature. Next, 5 g of powder was suspended in 100 mL of sterilized water at 65 °C for 1 h. After centrifugation at 15,000 rpm for 10 min, the supernatant was collected, added with 1% CaCl_2_, and incubated at 4 °C overnight to precipitate alginic acid. After centrifugation at 15,000 rpm for 10 min, the supernatant was collected, and 99% ethanol was added to achieve a final ethanol concentration of 30%. Subsequently, the supernatant was stored at 4 °C for 4 h and centrifuged again. Then, 99% ethanol was added to the supernatant to achieve a final concentration of 70% ethanol. Following overnight incubation, fucoidan was obtained after filtering the supernatant through a 0.45 μm nylon membrane.

### 4.3. FCC Purification

The separated fucoidan was suspended in 25 mL of distilled water. Then, 0.75 mL of 3.0 M HCL was added and mixed for 3 h with heating. After cooling to room temperature, centrifugation was performed at 5000 rpm for 10 min, and 1.0 M of NaOH was added to the supernatant. Subsequently, 100 mL of ethanol was added to the mixture, which was then centrifuged again at 5000 rpm for 10 min; the precipitate was dissolved in the distilled water and freeze-dried. All chemicals used for FCC extraction, including 99% ethanol (CAS No.: 64-17-5), were purchased from Sigma-Aldrich (St. Louis, MO, USA). The weight average molecular weight of FCC was analyzed using high-performance liquid chromatography (HPLC; Agilent Technologies, Santa Clara, CA, USA) combined with gel permeation chromatography.

### 4.4. Monosaccharide Composition

High-performance liquid chromatography (HPLC) was performed to detect monosaccharides present in fucoidan. Isolated fucoidan (60 mg) was mixed with 2 M of trifluoroacetic acid (TFA) and incubated at 120 °C for 5 h. After evaporating TFA, the sample was suspended in 100 μL of distilled water. After filtering through a 0.20 μm filter, the sample was injected into the HPLC system.

### 4.5. Reagents

Fucoidan from *Fucos vesiculosus* (F8190), LPS, and Dex were purchased from Sigma-Aldrich (St. Louis, MO, USA). Fluorescently labeled Brilliant Violet 605™ anti-mouse CD80 (Clone no. 16-10A1, cat no. 104729), Brilliant Violet 785™ anti-mouse CD11c (Clone no. N418, cat no. 117335), PE/Cyanine7 anti-mouse CD86 (Clone no. GL-1, cat no. 105013), PerCP/Cyanine5.5 anti-mouse H-2Kb (Clone no. AF6-88.5, cat no. 116515), APC anti-mouse CD40 (Clone no. 3/23, cat no. 124611), and PerCP anti-mouse I-A/I-E (Clone no. M5/114.15.2, cat no. 107623) antibodies were purchased from BioLegend (San Diego, CA, USA).

### 4.6. Animals

C57BL/6 mice were obtained from the Shanghai Public Health Clinical Center (SPHCC). The mice were reared in pathogen-free conditions at 20–22 °C and 50–60% humidity and provided a standard rodent diet and water. Experiments were performed following the guidelines of the Institutional Animal Care and Use Committee at SPHCC. The animal protocol was approved by the Ethics of Animal Experiments Committee of SPHCC (Protocol number: 2021-A070-01). CO_2_ inhalation euthanasia was used to euthanize the mice.

### 4.7. BMDC Generation

Bone marrow was extracted from the femur and calf bones of C57BL/6 mice. The bone marrow within the bone was separated using a 27 G needle and syringe. A single-cell bone marrow suspension was prepared and centrifuged at 1700 rpm for 7 min. Red blood cell (RBC) lysis buffer (BioLegend) was added to the pellet to remove RBCs, followed by washing with PBS. Cells (2 × 10^5^ cells/well) were seeded into 24 well plates and incubated with granulocyte macrophage-colony stimulating factor and IL-4 (100 ng/mL; both from BioLegend) for six days. BMDCs were confirmed by detecting the expression of CD11c six days after differentiation induction, with the presence of more than 90% of CD11c-positive cells confirmed before the next experiment.

### 4.8. Treatment and Analysis of BMDCs

Differentiation-induced BMDCs were treated with 0, 10, 20, and 50 μg/mL of FCC, and 1 h later, the cells were also treated with 20 ng/mL LPS. After 24 h of LPS stimulation, changes in BMDC morphology, expression of surface activity markers, and pro-inflammatory cytokine production were analyzed.

### 4.9. Flow Cytometric Analysis

BMDCs were incubated with isotype control and Fc receptor-blocking antibodies for 30 min to block non-specific antibody binding. Next, to confirm the activity of BMDCs, a fluorescent monoclonal antibody was added without washing. 4′,6-Diamidino-2-phenylindole staining was performed to exclude dead cells from the analysis, given the possibility of non-specific fluorescence expression. Subsequently, the cells were analyzed using flow cytometry (Becton Dickinson, Franklin Lakes, NJ, USA).

### 4.10. Enzyme-Linked Immunosorbent Assay (ELISA)

The secretion of pro-inflammatory cytokines was measured in the BMDC culture medium treated with FCC and LPS. ELISA was performed according to the manufacturer’s instructions (BioLegend). Briefly, the capture antibody for the corresponding cytokine was coated on a 96-well plate 1 day before analysis, and non-specific binding was blocked for 2 h with a blocking buffer containing fetal bovine serum. The sample was loaded and incubated for 2 h. After washing five times, the detection antibody was added, followed by incubation for 1 h. After washing again five times, HRP-conjugated secondary antibody was added and incubated for 30 min. The color development was monitored by adding a substrate and analyzed using a plate reader (Hangzhou Allsheng Instruments Co., Ltd., Hangzhou, China) after adding the stop solution.

### 4.11. LPS Injection and FCC Treatment

FCC (25, 50, or 100 mg/kg) was orally administered to C57BL/6 mice daily for 3 days. Six hours after the last administration, 100 μg/kg LPS was administered intraperitoneally. Twenty hours after LPS administration, BAL fluid and the lungs were harvested for additional experiments.

### 4.12. Hematoxylin and Eosin (H&E) Staining

After collecting BAL fluid, the lungs were flushed with 3.7% formaldehyde and then harvested. The lungs were fixed in 3.7% formaldehyde for 24 h, followed by dehydration using acetone and chloroform. The lung tissue was embedded in paraffin, sectioned into 5 μm thickness, and attached to a slide glass. After drying, the sections were rehydrated using xylene and observed under a microscope after H&E staining.

### 4.13. LPS-Induced Sepsis Model

C57BL/6 mice were administered FCC (0, 25, 50, or 100 mg/kg, dissolved in distilled water) orally, once daily for 3 days. Mice were assigned to six groups: a group treated with PBS alone, a group treated with LPS (10 mg/kg) alone, a group treated with FCC (50 mg/kg, daily for 3 days), and three groups treated with 25, 50, or 100 mg/kg FCC plus LPS. An additional group treated with Dex (5 mg/kg) was used as a positive control. LPS was administered intraperitoneally. Ten mice (*n* = 10) were tested in each group, and the survival rate was determined.

### 4.14. Statistical Analysis

All results are expressed as the mean and the standard error of the mean. Unless otherwise stated, experiments were repeated two times, and a total of six samples were analyzed (*n* = 6). Statistical significance was evaluated using one-way analysis of variance (ANOVA) and calculated using Tukey’s test. Values < 0.05 were deemed statistically significant.

## 5. Conclusions

In summary, we extracted and characterized FCC using an alcohol extraction method and HPLC and further evaluated its anti-inflammatory effect in BMDCs and LPS-induced mouse models of inflammation. The results provide evidence that FCC effectively suppresses LPS-induced inflammation in both BMDCs and mouse lung models. Furthermore, orally administered 50 mg/kg of FCC can suppress LPS-induced sepsis, which showed effects comparable to intraperitoneal administration of 5 mg/kg of Dex, a well-known anti-inflammatory agent. These findings confirm that FCC could be developed as a health supplement with anti-inflammatory properties.

## Figures and Tables

**Figure 1 marinedrugs-22-00401-f001:**
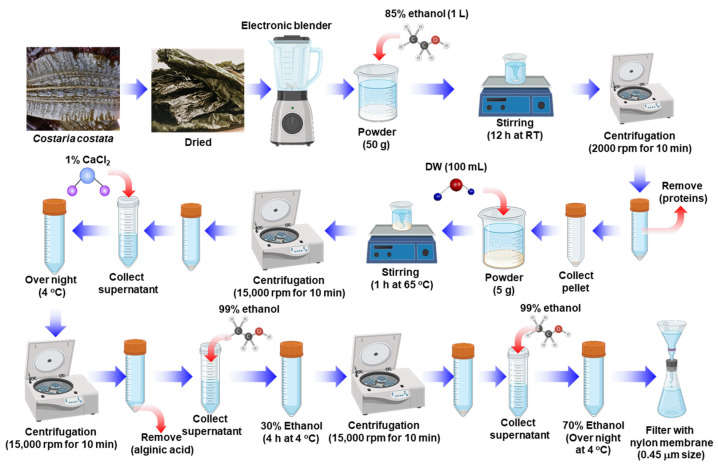
Schematic diagram illustrating the isolation procedure for fucoidan from *Costaria costata* (FCC).

**Figure 2 marinedrugs-22-00401-f002:**
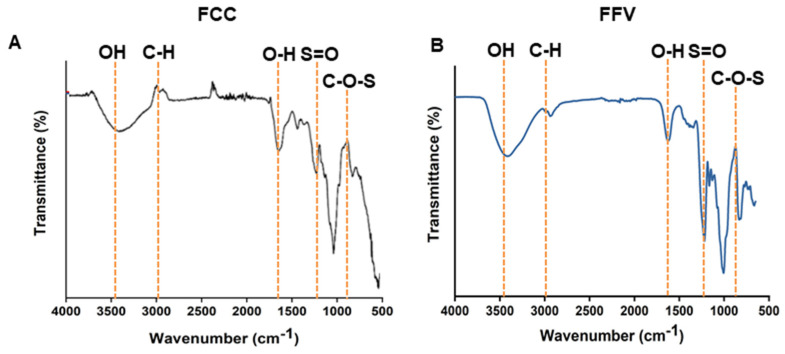
Fourier transform infrared (FT-IR) spectrometer images of FCC. FT-IR data of FCC (**A**) and Fucoidn from *F. vesiculosis* (FFV) (**B**) were shown. Specific chemical bonds are indicated. FCC, fucoidan extracted from *Costaria costata*.

**Figure 3 marinedrugs-22-00401-f003:**
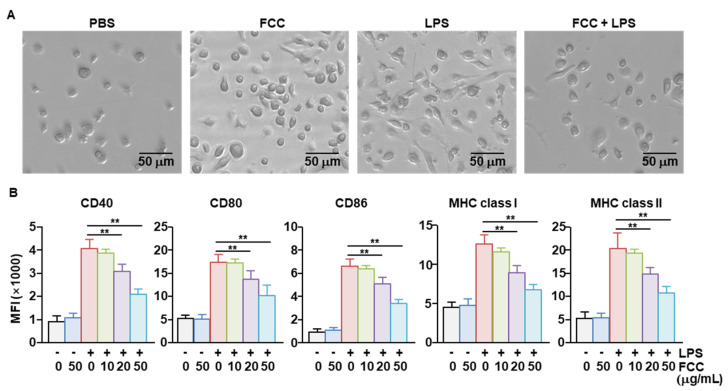
FCC inhibits LPS-induced activation of BMDCs. BMDCs were pretreated with the indicated dose of FCC. One hour after FCC treatment, the cells were stimulated with LPS for 24 h. (**A**) Morphological changes in BMDCs were examined using a microscope (50 μg/mL of FCC and 20 ng/mL of LPS). (**B**) Surface activation marker expression in BMDCs was analyzed using flow cytometry. The −/+ on the abscissa axis indicates the presence or absence of LPS, and the numbers below indicate the concentration of FCC. All data are averages or representative from six independent samples (*n* = 6), ** *p* < 0.01. BMDCs, bone marrow-derived dendritic cells; FCC, fucoidan extracted from *Costaria costata*; LPS, lipopolysaccharide.

**Figure 4 marinedrugs-22-00401-f004:**
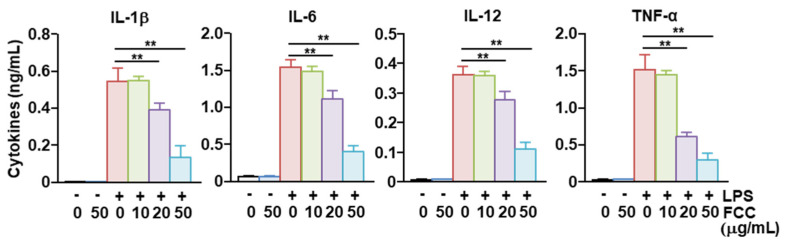
FCC inhibits LPS-induced pro-inflammatory cytokine secretion in BMDCs. As shown in Figure 3, BMDCs were treated with LPS after pretreatment with FCC. The concentrations of IL-1β, IL-6, IL-12, and TNF-α secreted from the BMDC culture medium were analyzed using ELISA. The −/+ on the abscissa axis indicates the presence or absence of LPS, and the numbers below indicate the concentration of FCC. All data are averages or representative from six independent samples (*n* = 6), ** *p* < 0.01. BMDCs, bone marrow-derived dendritic cells; ELISA, enzyme-linked immunosorbent assay; FCC, fucoidan extracted from *Costaria costata*; LPS, lipopolysaccharide; IL, interleukin; and TNF-α, tumor necrosis factor-α.

**Figure 5 marinedrugs-22-00401-f005:**
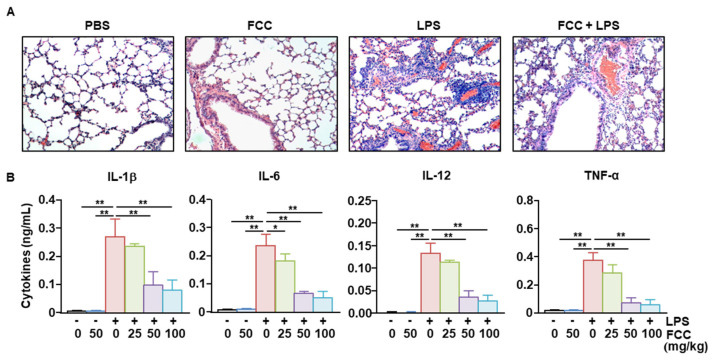
FCC inhibits LPS-induced acute lung inflammation. C57BL/6 mice were orally administered the indicated dose of FCC daily for 3 days. Six hours later, LPS (100 μg/kg) was administered intraperitoneally. (**A**) H&E staining of the lung after treatment with 50 mg/kg of FCC and 100 μg/kg of LPS. (**B**) Pro-inflammatory cytokines secreted in BAL fluid were measured using ELISA. The −/+ on the abscissa axis indicates the presence or absence of LPS, and the numbers below indicate the concentration of FCC. Results are expressed as the representative or average of six mice. (*n* = 6), * *p* < 0.05, ** *p* < 0.01. BAL, bronchoalveolar lavage; ELISA, enzyme-linked immunosorbent assay; FCC, fucoidan extracted from *Costaria costata*; H&E, hematoxylin, and eosin; LPS, lipopolysaccharide; IL, interleukin; PBS, phosphate-buffered saline; and TNF-α, tumor necrosis factor-α.

**Figure 6 marinedrugs-22-00401-f006:**
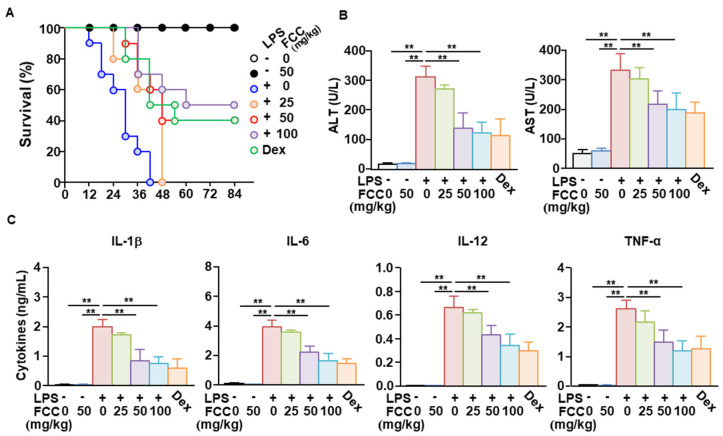
Oral treatment with FCC inhibits LPS-induced sepsis in mice. C57BL/6 mice were orally administered the indicated dose of FCC daily intervals for 3 days. Twenty-four hours after FCC treatment, LPS (10 mg/kg) was administered intraperitoneally. (**A**) Survival rates of mice for each group (*n* = 10). (**B**) Serum levels of alanine transaminase (ALT) and aspartate transaminase (AST). (**C**) Serum concentration of indicated pro-inflammatory cytokines. (**B**,**C**) The −/+ on the abscissa axis indicates the presence or absence of LPS, and the numbers below indicate the concentration of FCC. Dexamethasone (Dex; 5 mg/kg) was used as a positive control. Results are expressed as the average of six mice (*n* = 6), ** *p* < 0.01. FCC, fucoidan extracted from *Costaria costata*; LPS, lipopolysaccharide; IL, interleukin; PBS, phosphate-buffered saline; TNF-α, tumor necrosis factor-α.

**Table 1 marinedrugs-22-00401-t001:** Monosaccharide composition of fucoidans.

Polysaccharide Source	Composition of Neutral Sugar ^a^					Uronic Acid ^b^	SO_4_^2− c^	M.W. (KDa)
	Fucose	Xylose	Glucose	Mannose	Galactose			
Fucoidan from *Fucus vesiculosus* (%)	29.8 ± 1.3	2.4 ± 0.2	0.9 ± 0.1	1.2 ± 0.1	3.2 ± 0.1	5.5 ± 0.21	24.5 ± 1.2	
Fucoidan from *Costaria costata* (%)	25.4 ± 2.1	2.3 ± 0.3	3.1 ± 0.2	5.2 ± 0.3	4.1 ± 0.2	3.8 ± 0.4	26.7 ± 0.9	420

^a^ Determined by HPLC after acidic hydrolysis [34]. ^b^ Determined by carbazole method and calculated as glucuronic acid equivalent [34]. ^c^ Determined by turbidimetric assay after acidic hydrolysis [35]. The mean and standard errors of the results obtained through three repetitions are shown.

## Data Availability

The data discussed in this study are available on request from the corresponding author.

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
