# Peer review of "Anti-Inflammatory Effect of Fucoidan from Costaria costata Inhibited Lipopolysaccharide-Induced Inflammation in Mice"

_marinedrugs, 2024, doi:10.3390/md22090401_

Round 1
Reviewer 1 Report
Comments and Suggestions for Authors
Wei Zhang et al have submitted the manuscript about Anti-inflammatory Effect of Fucoidan from Costaria costata by inhibition of Lipopolysaccharide-induced Inflammation in Mice. After close evaluation of the amnuscript I have next recommendations:
1.In abstract authors provide information that have '...examined ... immune-modulating ability', while in the title 'Anti-inflammatory Effect of Fucoidan' was postulated.
2. In Introduction: it is worthy to mention that anti-inflammatory activities of fucoidans were studied both in vitro and in vivo. IRecently different mechanisms such as inhibition of COX-1 and COX-2, MAPK p38, hyaluronidase, protein denaturation, HRBC membrane stabilization, etc. were reported for fucoidans from brown seaweeds. In vivo, fucoidan based formulations dose-dependently inhibited carrageenan-induced edema and ameliorated mechanical allodynia in rats.
3. Table 1: authors provide Monosaccharide composition of fucoidans incl. composition of fucoidan from F. vesiculosus. However, no information about source of fucoidan from F. vesiculosus was provided.
4. No statistics was indicated in Table 1. How many samples of fucoidan from Costaria costata? It seems that only one saple of fucoidan was obtained and studied. The reliability of the results without statistical data is questionable.
5. Usually, sulfates content and molecular weight indicated for fucoidans. Please update Table 1.
6. Fig. 3 and 4 please provide legend for abscissa axis.
7. In Sect. 2.5 authors have performed single dose experiment in vivo. The scientific value of single dose experiment is very low.
8. Please indicate in which solvent FCC was dissolved befor per oral administration.
9. In Sect. 4.1. please provide information at which reproductive phase seaweeds were collected. Who have identified seaweed samples, please indicate voucher of specimens numbe.
10. The conclusion must be based on properly conducted experiments. However only single dose of FCC was used by authors in vivo. Authors must perform multi dose study using proper positive control and must compare efficacy of FCC with the efficacy of positive control.
Comments on the Quality of English Language
Minor language correction by professionals will be suitable.
Author Response
Reviewer 1.
Wei Zhang et al have submitted the manuscript about Anti-inflammatory Effect of Fucoidan from Costaria costata by inhibition of Lipopolysaccharide-induced Inflammation in Mice. After close evaluation of the amnuscript I have next recommendations:
Answer (A): Thank you for your important comments. I have added experiments and revised the text with answers to all your questions based on your comments. We would like to once again ask you to review our paper.
- In abstract authors provide information that have '...examined ... immune-modulating ability', while in the title 'Anti-inflammatory Effect of Fucoidan' was postulated.
A: Thanks for the reviewer's comments. As the reviewer pointed out, ‘immune-modulating’ was changed to ‘anti-inflammatory’.
- In Introduction: it is worthy to mention that anti-inflammatory activities of fucoidans were studied both in vitro and in vivo. Recently different mechanisms such as inhibition of COX-1 and COX-2, MAPK p38, hyaluronidase, protein denaturation, RBC membrane stabilization, etc. were reported for fucoidans from brown seaweeds. In vivo, fucoidan based formulations dose-dependently inhibited carrageenan-induced edema and ameliorated mechanical allodynia in rats.
A: The introduction section has been revised, including references pointed out by the reviewer.
- Table 1: authors provide Monosaccharide composition of fucoidans incl. composition of fucoidan from F. vesiculosus. However, no information about source of fucoidan from F. vesiculosus was provided.
A: Fucoidan from F. vesiculosus was purchased from sigma-aldrich. This information has been added to the Reagents section.
- No statistics was indicated in Table 1. How many samples of fucoidan from Costaria costata? It seems that only one saple of fucoidan was obtained and studied. The reliability of the results without statistical data is questionable.
A: The errors were reported in the table by reanalyzing three samples during the revision.
- Usually, sulfates content and molecular weight indicated for fucoidans. Please update Table 1.
A: We supplemented Table 1 by additionally analyzing the contents of Uronic Acid and SO42-.
- Fig. 3 and 4 please provide legend for abscissa axis.
A: We have now revised the figure legends.
- In Sect. 2.5 authors have performed single dose experiment in vivo. The scientific value of single dose experiment is very low.
A: The data in Figures 5 and 6 were revised by experiments at concentration dependent effect of FCC.
- Please indicate in which solvent FCC was dissolved befor per oral administration.
A: FCC was dissolved in distilled water and administered orally, as described in the Methods section.
- In Sect. 4.1. please provide information at which reproductive phase seaweeds were collected. Who have identified seaweed samples, please indicate voucher of specimens numbe.
A: We have revised the material section.
- The conclusion must be based on properly conducted experiments. However only single dose of FCC was used by authors in vivo. Authors must perform multi dose study using proper positive control and must compare efficacy of FCC with the efficacy of positive control.
A: Thanks again for the reviewer's valuable comments. In accordance with the reviewer's comments, we examined the anti-inflammatory effect of FCC at three concentrations against LPS. In addition, dexamethasone, a well-known anti-inflammatory agent, was added as a positive control group and its efficacy with FCC was compared. Accordingly, Figures 5 and 6 were completely re-experimented and revised, and the corresponding figure legend and result section were also revised.

Reviewer 2 Report
Comments and Suggestions for Authors
I have read the manuscript and have several questions and recommendations. 1. In section 4.1, it is necessary to indicate the time of collection of the algae. Specify who identified the sample and the location of storage of the herbarium specimen.
2. The description of the technology in Fig. 1 is confusing. Please explain in the figure what is discarded at each stage. Otherwise, it is clear from the figure that 99% ethanol is used for extraction.
3. Specify the manufacturer and catalog number of 99% ethanol.
4. The authors compare their fucoidan sample with fucoidan from Fucus vesiculosis. Specify the origin of this sample and its expiration date (if it was commercial).
5. It has been previously shown that In Vitro Anti-Inflammatory Activities of Fucoidans from Five Species of Brown Seaweeds was concentration-dependent and strongly correlated with the fucose content and moderate with sulfate content. The purified fucoidan from Fucus vesiculosis showed the most promising activity, exceeding the reference drug diclofenac sodium. Pharmacological data without comparison with reference drugs have low scientific significance. Please supplement your studies with a reference drug (a drug widely used to treat inflammation). Compare the data of the reference drug and your fucoidan. Compare the data of your drug with the literature.
6. Numerous studies show that molecular weight, sulfate content, fucose content, and polyphenols may contribute to these activities. Please provide data on the definition of molecular weight, sulfate content, and polyphenols.
7. In Figure 2, provide the FTIR data for fucoidan from Fucus vesiculosis. Please compare the FTIR of fucoidan from Costaria costata with the FTIR of referent sample.
8. The Discussion section needs to be rewritten. It should describe the authors' data in comparison with those obtained previously. For example, a comparison of the monosaccharide composition with that previously determined for Costaria costata, etc.
9. Conclusions need to be rewritten, since there are no data for the comparison drug and it is impossible to draw a conclusion about the activity.
Author Response
Reviewer 2
I have read the manuscript and have several questions and recommendations.
Answer (A): Thank you for your important comments. I have added experiments and revised the text with answers to all your questions based on your comments. We would like to once again ask you to review our paper.
- In section 4.1, it is necessary to indicate the time of collection of the algae. Specify who identified the sample and the location of storage of the herbarium specimen.
A: We have revised the material section.
- The description of the technology in Fig. 1 is confusing. Please explain in the figure what is discarded at each stage. Otherwise, it is clear from the figure that 99% ethanol is used for extraction.
A: Thank you for your comments. We are sorry for the confusing description of the isolation procedure for FCC and the wrong concentration of ethanol in Fig.1, we used 85% ethanol to remove proteins and the pellet was collected after centrifugation. We have now revised Fig.1 and method section for extraction of FCC.
- Specify the manufacturer and catalog number of 99% ethanol.
A: The 99% ethanol was purchased from Sigma-Aldrich.
- The authors compare their fucoidan sample with fucoidan from Fucus vesiculosis.Specify the origin of this sample and its expiration date (if it was commercial).
A: Fucoidan from F. vesiculosus was purchased from sigma-aldrich. This information has been added to the Reagents section.
- It has been previously shown that In Vitro Anti-Inflammatory Activities of Fucoidans from Five Species of Brown Seaweeds was concentration-dependent and strongly correlated with the fucose content and moderate with sulfate content. The purified fucoidan from Fucus vesiculosis showed the most promising activity, exceeding the reference drug diclofenac sodium. Pharmacological data without comparison with reference drugs have low scientific significance. Please supplement your studies with a reference drug (a drug widely used to treat inflammation). Compare the data of the reference drug and your fucoidan. Compare the data of your drug with the literature.
A: We have conducted various studies on Fucoidan, but most of them have confirmed functions related to immune activity. In particular, in vitro and in vivo experiments, the immune activation ability of fucoidan was able to induce an anticancer effect {Zhang, 2015 #30;Jin, 2014 #52;Jin, 2015 #53;An, 2022 #54; Zhang, 2021 #55;Park, 2020 #56;Zhang, 2021 #57}, whereas the FCC extracted this time showed an anti-inflammatory effect. Therefore, we could not find a suitable fucoidan to compare with the fucoidan used in this study, and therefore, we used Dexamethasone, a well-known anti-inflammatory agent, as a positive control and compared its effects with FCC.
- Numerous studies show that molecular weight, sulfate content, fucose content, and polyphenols may contribute to these activities. Please provide data on the definition of molecular weight, sulfate content, and polyphenols.
A: We have updated Table 1 with information on FCC by adding all the experiments as much as we can. As well as the method section was updated.
- In Figure 2, provide the FTIR data for fucoidan from Fucus vesiculosis. Please compare the FTIR of fucoidan from Costaria costata with the FTIR of referent sample.
A: We have added FT-IR data for Fucoidan from Fucus vesiculosis in Figure 2B.
- The Discussion section needs to be rewritten. It should describe the authors' data in comparison with those obtained previously. For example, a comparison of the monosaccharide composition with that previously determined for Costaria costata, etc.
A: The discussion section was revised by comparing the polysaccharide content and anti-inflammatory effects of FCC.
- Conclusions need to be rewritten, since there are no data for the comparison drug and it is impossible to draw a conclusion about the activity.
A: We revised this paper by performing an in vivo experiment using dexamethasone (Dex), a well-known anti-inflammatory agent, as a positive control and added a corresponding explanation to the experimental results. This is expected to further support the conclusion about the anti-inflammatory effect of FCC.

Reviewer 3 Report
Comments and Suggestions for Authors
In this manuscript entitled "Anti-infection effect of fucoidan from Costa Rica inhibited lipolysis in mice", the mechanism of fucoidan extracted from Costaria costata in treating inflammation was fully discussed, and its anti-inflammatory effect was confirmed, and its possibility as a health supplement was indicated. This study has further explored the pathogenesis of inflammation and explored how to treat inflammation, and provided a new way of thinking for human beings to explore ways to treat inflammation. The following are some questions about the manuscript and the places that can be further improved in the manuscript:
1. In the Introduction part, we should explain the current research background and emphasize the innovation of this research, so that readers can understand it intuitively.
2. In part 4.9, how is the injection amount of FCC determined, and what is the basis?
3. In the discussion part, the practical significance of this study can be explained, as well as the innovations or outstanding characteristics in curative effect compared with the existing drugs.
4. In part 8.5, the author chooses to use ethanol extraction method to extract fucoidan. Is it to quote other people's literature and modify it? If so, the cited literature should be added.
Author Response
Reviewer 3.
In this manuscript entitled "Anti-infection effect of fucoidan from Costa Rica inhibited lipolysis in mice", the mechanism of fucoidan extracted from Costaria costata in treating inflammation was fully discussed, and its anti-inflammatory effect was confirmed, and its possibility as a health supplement was indicated. This study has further explored the pathogenesis of inflammation and explored how to treat inflammation, and provided a new way of thinking for human beings to explore ways to treat inflammation. The following are some questions about the manuscript and the places that can be further improved in the manuscript:
- In the Introduction part, we should explain the current research background and emphasize the innovation of this research, so that readers can understand it intuitively.
Answer (A): Thank you for your important comments. I have added experiments and revised the text with answers to all your questions based on your comments. We would like to once again ask you to review our paper.
- In part 4.9, how is the injection amount of FCC determined, and what is the basis?
A: We examine the anti-inflammatory effect of FCC in a concentration-dependent manner and added it to the revised version. The anti-inflammatory effect of 50 mg/kg FCC was stronger than that of 25 mg/kg, and this was not significantly different from that of 100 mg/kg FCC, therefore 50 mg/kg was determined as the optimal concentration.
- In the discussion part, the practical significance of this study can be explained, as well as the innovations or outstanding characteristics in curative effect compared with the existing drugs.
A: We revised the discussion section to further emphasize the importance of the implications of the results of this study.
- In part 8.5, the author chooses to use ethanol extraction method to extract fucoidan. Is it to quote other people's literature and modify it? If so, the cited literature should be added.
A: A paper describing the fucoidan extraction method was cited.

Round 2
Reviewer 1 Report
Comments and Suggestions for Authors
Authors have revised and updated the manuscript. However, I have some additional recommendations.
1. Dose-dependent inhibition of carrageenan-induced edema and amelioration of mechanical allodynia in rats was described in one article dedicated to Formulation, Optimization and In Vivo Evaluation of Fucoidan-Based Cream with Anti-Inflammatory Properties
2. Please add molecular weight in Table 1 as recommended in previous round.
3. In Sect. 4.1. please provide information who have identified seaweed samples,and please indicate voucher of specimens number
4. In conclusion: please update the phrase "Furthermore, orally administered FCC can suppress LPS-induced sepsis, which showed effects comparable to those of dex, " with information about doses of FCC and dex. It will ubderline of effects were comparable.
Author Response
- Dose-dependent inhibition of carrageenan-induced edema and amelioration of mechanical allodynia in rats was described in one article dedicated to Formulation, Optimization and In Vivo Evaluation of Fucoidan-Based Cream with Anti-Inflammatory Properties
Answer (A): I apologize for this mistake. The paper on the subject was cited again and the text was revised.
- Please add molecular weight in Table 1 as recommended in previous round.
A: We have now revised MW in Table 1.
- In Sect. 4.1. please provide information who have identified seaweed samples,and please indicate voucher of specimens number
A: Professor Sang-Kwan Yoo of the Department of Marine Food Science and Technology, Gangneung-Wonju National University in Korea confirmed Costaria costata. The specimens are stored in the Shanghai public health clinical center, with the voucher number SPH-DA-1.
- In conclusion: please update the phrase "Furthermore, orally administered FCC can suppress LPS-induced sepsis, which showed effects comparable to those of dex," with information about doses of FCC and dex. It will ubderline of effects were comparable.
A: The dosage and route of administration of FCC and Dex are specified in the conclusion section.

Reviewer 2 Report
Comments and Suggestions for Authors
The authors partially answered my questions, in particular, please indicate the voucher number, who personally identified and the storage location of the herbarium sample of algae.
Author Response
The authors partially answered my questions, in particular, please indicate the voucher number, who personally identified and the storage location of the herbarium sample of algae.
Answer: Professor Sang-Kwan Yoo of the Department of Marine Food Science and Technology, Gangneung-Wonju National University in Korea confirmed Costaria costata. The specimens are stored in the Shanghai public health clinical center, with the Voucher number SPH-DA-1.

Round 3
Reviewer 1 Report
Comments and Suggestions for Authors
Authors have revised the manuscript, however, two minor points require attention..
1. Please delete '{Hu, 201 #48;Lukova, 2023 #47} in lines 47-48.
2. In Reference list: reference 21 is not complete.